# Recent Advances in Microfluidic-Based Extracellular Vesicle Analysis

**DOI:** 10.3390/mi15050630

**Published:** 2024-05-08

**Authors:** Jiming Chen, Meiyu Zheng, Qiaoling Xiao, Hui Wang, Caixing Chi, Tahui Lin, Yulin Wang, Xue Yi, Lin Zhu

**Affiliations:** 1Department of Basic Medicine, Xiamen Medical College, Xiamen 361023, China; jimmychen131@163.com (J.C.); zhengmeiyufjmu@163.com (M.Z.); 202200010531@xmmc.edu.cn (Q.X.); lyuxi8549@gmail.com (H.W.); ccx20221214@163.com (C.C.); tahui.lin@foxmail.com (T.L.); 202200010532@xmmc.edu.cn (Y.W.); 2Key Laboratory of Functional and Clinical Translational Medicine, Fujian Province University, Xiamen 361023, China; 3Institute of Respiratory Diseases, Xiamen Medical College, Xiamen 361023, China

**Keywords:** extracellular vesicles, microfluidics, EV isolation, EV detection

## Abstract

Extracellular vesicles (EVs) serve as vital messengers, facilitating communication between cells, and exhibit tremendous potential in the diagnosis and treatment of diseases. However, conventional EV isolation methods are labor-intensive, and they harvest EVs with low purity and compromised recovery. In addition, the drawbacks, such as the limited sensitivity and specificity of traditional EV analysis methods, hinder the application of EVs in clinical use. Therefore, it is urgent to develop effective and standardized methods for isolating and detecting EVs. Microfluidics technology is a powerful and rapidly developing technology that has been introduced as a potential solution for the above bottlenecks. It holds the advantages of high integration, short analysis time, and low consumption of samples and reagents. In this review, we summarize the traditional techniques alongside microfluidic-based methodologies for the isolation and detection of EVs. We emphasize the distinct advantages of microfluidic technology in enhancing the capture efficiency and precise targeting of extracellular vesicles (EVs). We also explore its analytical role in targeted detection. Furthermore, this review highlights the transformative impact of microfluidic technology on EV analysis, with the potential to achieve automated and high-throughput EV detection in clinical samples.

## 1. Introduction

Extracellular vesicles (EVs) are membranous particles enclosed by lipid bilayers with diameters ranging from 50 nm to 1 μm, derived from almost all cells [1]. By inheriting abundant signaling molecules from donor cells, such as nucleic acids and proteins, EVs play a role in intercellular communication and are associated with disease progression [2,3,4]. EVs can be classified into several subgroups based on size and generation mechanism, including microvesicles, microparticles, and exosomes [3,5]. Among these subgroups, exosomes are microvesicles ranging in size from 50 to 150 nm, with an average of ~100 nm [3]. Multivesicular bodies (MVBs) contain multiple intraluminal vesicles (ILVs), which fuse with the plasma membrane to release ILVs as exosomes [6]. Therefore, exosomes contain proteins of the cell surface and soluble proteins linked to the extracellular environment [7,8]. Efficient isolation and reliable analysis of EVs are important prerequisites for the clinical application of EVs. Conventional techniques for isolating EVs, such as ultracentrifugation, ultrafiltration, particle size exclusion chromatography, and polymer precipitation, are time-consuming and inefficient [9]. Microfluidic technology achieves efficient EV isolation and detection with only a very small amount (10^−3^ to 10^−12^ microliters) of liquid specimens and reagents and has the advantages of integration and high throughput [10].

The high throughput capability of microfluidics-based capturing and detection technologies offers a significant advantage in the field of EV analysis. Through precise manipulation of fluids at the microscale level, these technologies enable rapid and efficient isolation and detection of EVs from complex biological samples. This not only expedites the process of EV analysis but also allows for the examination of a large number of samples in a relatively short period, facilitating high-throughput screening and analysis [11]. Furthermore, microfluidic-based EV capture and detection technologies support standardization of experimental protocols and assay conditions, promoting consistency and reproducibility across studies. Through precise control of fluidic parameters, such as flow rate, shear stress, and incubation time, microfluidic platforms enable reproducible isolation and characterization of EVs with minimal batch-to-batch variability. Furthermore, the integration of quality control features, such as on-chip calibration and reference standards, facilitates accurate quantification and comparison of EVs between different experiments and laboratories [12]. Therefore, the standardization of microfluidics-based platforms ensures reproducibility and reliability, making them valuable tools for EV research and clinical applications.

In this review, we will explore the latest developments in microfluidic-based EV isolation and analysis, as well as discuss the challenges and future directions of this rapidly evolving field.

## 2. Microfluidic-Based EV Isolation Strategies

### 2.1. Label-Free Microfluidic Isolation

Label-free isolation strategies depend on the physical characteristics of EVs, such as their size, density, and deformability, to separate them from other components present in bodily fluids. Conventional label-free methods, such as ultrafiltration (UF), ultracentrifugation (UC), and size-exclusion chromatography (SEC), usually require large sample volumes. UC differentiates EVs and other components based on size and density and has been recognized as one of the most commonly used EV isolation methods [13,14]. However, the long UC period might result in the coprecipitation of protein aggregates. In addition, the quality of EVs cannot be guaranteed due to repeated centrifugation processes and excessive centrifugal force [15]. UF adopts membrane filters with suitable pore sizes to purify EVs from other particles [11]. Regrettably, the accumulation of debris on the filter membranes will decrease the efficiency of EV isolation and shorten the lifespan of the membranes [16]. SEC isolates particles of different sizes based on their different flow rates in a column filled with porous beads. As a mild EV isolation method, SEC can obtain EVs with relatively complete structure and function [17,18]. However, the tedious procedure, low throughput, and recovery of SEC limit its wide application. 

Holding the properties of easy integration, high-throughput, and low sample consumption, microfluidics technology provides a variety of strategies for efficient label-free EV isolation [19,20,21]. For example, Liang et al. proposed a double-filtration microfluidic device to isolate EVs with a size range of 30–200 nm [22]. Li et al. integrated filtration and microfluidics technology to design cascaded microfluidic circuits for preprogrammed, clog-free, and gentle isolation of EVs directly from blood within 30 min (Figure 1a) [23]. The problems of filter fouling and particle aggregation were solved by the pulsatile flows generated by the porous membrane, which lift particles away from the membrane. Researchers engineered an automated centrifugal microfluidic disc featuring membranes with specific functionalities (Exo-CMDS) for exosome isolation (Figure 1b) [24]. As a one-step method, Exo-CMDS enriched exosomes with an optimal exosomal concentration of 5.1 × 10^9^ particles/mL from a small amount of blood samples (<300 μL) in 8 min. The above methods required tedious fabrication of microfluidic chips, which increased the uncertainty of trials. Viscoelastic microfluidic systems manipulate viscoelastic fluids based on their viscous and elastic characteristics under deformation without the complex design of microfluidic chips and have been used for micro-/nano-sized particles. Asghari et al. developed sheathless oscillatory viscoelastic microfluidics for focusing and separating EVs [25]. Liu et al. proposed a viscoelastic-based microfluidic system for direct label-free isolation of exosomes (Figure 1c) [26]. A simple microfluidic device with two inlets, three outlets, and a straight microchannel was designed for the viscoelastic separation of exosomes. The presence of oxyethylene (PEO) causes elastic lift forces, which drive EVs toward the microchannel centerline according to their sizes. Under the size cutoff of 200 nm, exosomes were separated from large EVs. In addition, Yang et al. demonstrated a self-adaptive virtual microchannel for nanoparticle enrichment and separation in a continuous manner (Figure 1d) [27]. A gigahertz bulk acoustic resonator, in combination with microfluidics, triggers and stabilizes acoustic waves and streams. This process forms a virtual channel whose diameter can self-adjust, ranging from dozens to a few micrometers. Using a customized arc-shaped resonator, exosomes from patient plasma were purified. The system is stable and has high automation potential because of the self-adaptive and contactless continuous speration mode. 

Deterministic lateral displacement (DLD)-patterned pillars separate particles based on the DLD critical diameter (D_c_) and have been used to isolate circulating tumor cells (CTCs), stem cells, bacteria, and EVs [28]. Zeming et al. captured EVs with 1 µm polymer beads, which increased the size of the polymer beads. As a result, the DLD lateral displacement of beads can be translated to the amount of EVs [29]. To separate nanoscale exosomes, Wunsch et al. designed nanoscale DLD arrays with gap sizes from 25 to 235 nm to isolate human-urine-derived exosomes with a single-particle resolution [30]. Smith et al. designed 1024 nanoscale DLD (nanoDLD) arrays in a microfluidic chip to isolate EVs from serum and urine samples with a yield of ~50% [31]. Moreover, RNA sequencing was produced after separating EVs from prostate cancer (PCa) patient samples, which has the potential to indicate the aggression of PCa. However, the conventional DLD methods need input pressure to drive EVs or beads through pillars in the microfluidic chip, which limits the application of DLD technology in EV isolation. To avoid input pressure, Hattori et al. proposed an electroosmotic flow-driven DLD strategy that used electroosmotic flow to drive fluids for EV separation in a continuous manner [32]. The property of being easy-to-operate makes it a promising solution for clinical diagnostic applications. 

Although label-free microfluidic isolation has the advantages of high throughput, rapidity, and cheapness, the low purity and inability to isolate EV subtypes limit its downstream applications.

### 2.2. Affinity-Based EV Isolation

Affinity-based isolation methods exploit the interaction between affinity ligands (antibodies, peptides, or aptamers) and receptors on EV membranes to isolate EVs specifically [33,34]. These ligands are typically modified on the surface of materials or interfaces, such as CIM^®^ CDI disks [35], magnetic beads [36], carbon cloth [37], graphene [38], or Ti2CTx MXene membranes [39]. The strength and duration of the interaction between affinity ligands and receptors on surfaces determine the capture efficiency of EVs. However, the compromised interaction between EVs and traditional interfaces limits their capture efficiency. The micro-/nanoscale channels in microfluidic chips can enhance the contact frequency between recognition ligands on the chips and molecules on EV membranes, thus increasing EV capture efficiency. For example, the herringbone microfluidic chip, comprising patterned microgrooves, enhances fluid mixing efficiency by manipulating flow states and forming helical motions [40]. Therefore, collision between biological targets and affinity-trapping substrates is improved, resulting in improved EV capture efficiency [41]. 

However, the near-surface hydrodynamic resistance decreases mass transfer in the microchannel [42]. To overcome this near-surface hydrodynamic resistance, Li et al. developed a 3D porous sponge microfluidic chip made by salt crystallization, which provided a high surface-to-volume ratio [8]. Moreover, researchers also developed a fluid nanoporous microinterface (FluidporeFace) in a herringbone microfluidic chip for the efficient capture of tumor-derived EVs (Figure 2a) [41]. Supported lipid bilayers (SLBs) were encapsulated on the nanoporous herringbone microstructures, which not only improved the mass transfer but also enabled multivalent recognition of aptamers, thus achieving a multi-scale enhanced affinity reaction. As a result, the affinity increased by ~83-fold compared with the nonfluid interface. In addition, they designed a microfluidic chip to create a dynamic multivalent magnetic interface, enhancing the kinetics and thermodynamics of biomolecular recognition for the efficient isolation of EVs derived from tumors (T-EVs) [43]. Utilizing magnetic and flow fields, this engineered interface achieved a harmonious balance of affinity, selectivity, reversibility, and extendibility. As a result, they achieved a high-throughput recovery of T-EVs, facilitating comprehensive protein profiling. However, when utilizing a single ligand for EV capture and another ligand for EV detection, it is challenging to eliminate the interference of free proteins and obtain the requisite subtypes of EVs, thereby rendering it inadequate for clinical applications. To eliminate interference from free proteins, Zhang et al. designed a microfluidic differentiation method that accurately captured PD-L1^+^ EVs [44]. PD-L1^+^ EVs were labeled with biotin using DNA computation, incorporating dual inputs of lipid probes and PD-L1 aptamers. Subsequently, these labeled EVs were captured with streptavidin-modified microfluidic chips selectively. 

Different subtypes of EVs represent distinct sources and diverse biological functions. Therefore, the analysis of EV subtypes is crucial for studying the biological mechanisms of EVs. On the contrary, all antibody-based enrichment systems are limited to highly specific isolation protocols, which result in partial EV loss due to differences in the expression of EV membrane surface proteins. Chen et al. proposed a novel herringbone microfluidics device that not only possessed the advantages of herringbone microfluidics but also incorporated aptamer-functionalized core–shell bar codes (AFCSBs) [40]. Because their antiopal hydrogel shells have abundant interconnecting pores, barcodes can provide a rich surface area for the anchoring of multiple DNA aptamers, enabling the specific capture of multiple tumor-derived exosomes. However, its essence lies in using one type of aptamer for molecular capture and another type for molecular discrimination, making it still unable to distinguish different EVs. In order to isolate subtypes of EVs, MUN et al. developed a microfluidic chip-based magnetically labeled exosome isolation system (MEIS-chip) that involved magnetic nanoclusters (MNCs) conjugated with CD63 and HER2 with different degrees of magnetization (CD63 conjugated with low-saturation magnetized MNCs, CD63-LMC, and HER2 conjugated with high-saturation magnetized MNCs, HER2-HMC) [47]. Common exosomes were captured by CD63-LMC, while exosomes with HER overexpression bound to both CD63-LMC and HER2-HMC simultaneously. This allows for the acquisition of varying degrees of magnetic particles and magnetic separation in the MEIS chip via a magnetic field, ultimately resulting in the separation of different EV populations. Moreover, Chen et al. designed a nanoscale cytometry platform called NanoEPIC to enable the collection of small EVs (sEVs) bearing four different expression levels of PD-L1 by labeling them with antibody-functionalized magnetic nanoparticles (MNPs) (Figure 2b) [45]. EVs with higher PD-L1 expression levels had greater lateral deflection towards the edge of the device in the microfluidic flow channel due to their increased magnetic susceptibility resulting from binding to more MNPs. This facilitated the separation of sEVs based on four levels of PD-L1 expression: negative, low (exoL), medium (exo-M), and high (exo-H). Lu et al. achieved the isolation of tumor PD-L1^+^ EVs and non-tumor PD-L1^+^ EVs through DNA logic-mediated double aptamer recognition and a tandem chip for the first time (Figure 2c) [46]. Tumor-derived EVs were identified by EpCAM and PD-L1 nucleic acid ligands, induci ng the “AND” logic operation, whereas non-tumor-derived PD-L1^+^ EVs only express PD-L1, thus invoking the “NOT” logic operation. These two independent outputs facilitated the separation of tumor- and non-tumor-origin PD-L1^+^ EVs through tandem microfluidics, respectively. Consequently, utilizing a streptavidin-functionalized microfluidic chip (T-Chip), only tumor-derived PD-L1^+^ EV populations can be isolated. After excluding tumor-derived PD-L1^+^ EVs, the remaining PD-L1^+^ EVs from normal cells can be captured through hybridization between the extension sequence on the PD-L1 probe and the corresponding cDNA modified on the second microfluidic chip (N-Chip).

Currently, microfluidic technology based on affinity separation has made tremendous advancements. There have been significant improvements in purification, capture efficiency, and subpopulation separation. Despite some progress in various studies, there are still certain limitations. For example, this method can suffer from non-specific binding to other entities present in the sample, such as proteins, lipoproteins, and cellular debris. This non-specific binding can lead to contamination and reduced purity in the isolated EV population [48]. Moreover, affinity-based EV isolation depends on the availability and specificity of surface markers for EV capture. However, not all EVs express the same surface markers, and the expression profile of EV surface markers can vary depending on cell type, physiological state, and environmental conditions. This limitation restricts the applicability of affinity-based methods, particularly when targeting specific subpopulations of EVs [14]. To effectively apply these technologies to clinical diagnosis and precision treatment, further innovation and improvement are still needed. In short, the efficient isolation of EVs is the premise for researching their biological function and clinical application. It is still necessary to develop new methods for the high-efficiency and high-purity isolation of EV subtypes.

In addition, here is a comparison presenting the effectiveness, efficiency, and practicality of traditional techniques versus microfluidic-based methodologies for EV analysis in Table 1 [49,50,51].

## 3. Microfluidic-Based EV Detection

### 3.1. Fluorescent Detection

Fluorescence technology combined with a microfluidic platform has been widely used in EV detection, which has the characteristics of fast response, good precision, and high sensitivity. After capturing EVs in microfluidic chips, membrane stains or fluorescent labeled antibodies are usually used to identify EVs [52]. For example, Kanwar et al. used a fluorescent carbanine dye (DiO) to stain the exosome membrane and counted the total number of exosomes captured on a microfluidic device (ExoChip) [12]. Antibodies against EV-specific biomarkers, such as CD63, CD9, and CD81, are used for EV identification and quantification [53]. Moreover, fluorescent-labeled antibodies against disease-related biomarkers on the EV membrane are usually used for quantitative and qualitative analysis of EV biomarkers, thus reflecting the progression of diseases. For example, Hisey et al. captured ovarian cancer exosomes in a herringbone groove microfluidic device and quantified EpCAM^+^ exosomes [54]. The quantitative results showed that EpCAM^+^ exosomes were related to HGSOC disease progression. After utilizing the integrated microfluidic exosome separation and detection system (EXID system), Lu et al. examined the abundance of exosomal PD-L1 [55]. Using the EXID system, a significant difference in fluorescence intensity was observed. The strategy had a limit of detection (LOD) of 10.76/µL, and the exosomal PD-L1 level reflected the sensibility for immune response. The conventional methods offer bulk information on proteins in EVs, which hardly enables absolute quantification. Liu et al. constructed “exosome-magnetic microbead-enzymatic reporter” complexes and encapsulated the complexes into droplets, which ensured a single complex was encapsulated in a droplet. As a result, cancer-specific exosomes were absolute counted with a low limit of detection of 10 exosomes µL^−1^ (Figure 3a) [56].

The heterogeneity of EVs challenges the acknowledgement of their biological significance and clinical application. To explore the heterogeneity of EVs, Zhang’s group developed a microfluidic chip featuring self-assembled 3D herringbone nanopatterns, enabling highly sensitive fluorescent detection of EV surface proteins [60]. The device was used to detect exosome subtypes expressing CD24, EpCAM, and FRalpha proteins in 2 μL plasma samples from 20 ovarian cancer patients and to suggest exosomal FRalpha as a promising biomarker in the early detection and monitoring of ovarian cancer progression. Furthermore, MMP14 on EVs holds potential for early detection and prognosis assessment in breast cancer metastasis [61]. The above microfluidic devices need cumbersome fabrication processes. Moreover, the interaction between proteins and antibodies may be limited due to the steric hindrance caused by the post-translational modification of proteins [62]. Sun’s group developed a microfluidic thermophoresis device that accumulated particles in a size-dependent manner and amplified fluorescence signals based on the different diffusion rates of particles in a nonuniform temperature field [63]. Seven fluorescently labeled aptamers targeting different epitopes were employed for the subtyping analysis of EVs. This method demonstrated a sensitivity and specificity of 95% and 100%, respectively, in cancer detection, and it can also be utilized for cancer classification. Furthermore, the method is applicable for the proteomic analysis of EVs in breast cancer, enabling the identification of metastatic breast cancer, monitoring treatment responses, and predicting patients’ progression-free survival rates [64].

In addition to phenotypic heterogeneity, the tracing of EV origins is also particularly important. It can accurately detect and monitor the progression of diseases. Sun’s group utilized microfluidic thermophoresis devices for the specific detection of tumor-derived EVs, achieving an accuracy of 97% [65]. Similarly, a rapid and non-invasive diagnostic assay, named the one-step thermophoretic AND gate operation (Tango), has been developed for precise identification of prostate cancer (PCa)-derived EVs directly in serum samples within 15 min. This method demonstrated an impressive overall accuracy of 91% in discerning PCa from benign prostatic hyperplasia (BPH) using a streamlined single-step format. This innovative technique holds tremendous potential for the swift and non-invasive diagnosis of cancers (Figure 3b) [57]. 

In addition to proteins on the EV surface, EV nucleic acids are also important biomarkers and therapeutic targets for diseases [66]. Shao et al. developed a microfluidic chip to capture exosomes and analyze the mRNAs of MGMT and APNG in enriched tumor exosomes [67]. This strategy has the potential to predict the drug responses of GBM patients. Sun’s group demonstrated a thermal swim sensor (TSN) employing nanoflares for the in situ detection of exosomal miRNA, eliminating the need for RNA extraction or target amplification. Through the thermophoretic accumulation of nanoflare-treated exosomes, a heightened fluorescent signal was produced upon binding with exosome miRNA, facilitating the direct quantitative assessment of exosome miRNA [68]. Afterwards, they devised a DNA tetrahedron-based thermophoretic analysis (DTTA) for the in situ detection of mRNA in EVs, achieving remarkable sensitivity and specificity [69]. Recently, they developed a DTTA for highly sensitive and selective in situ detection of mature miRNA in EVs. This assay achieved a detection limit of 2.05 fM for mature miRNA in EVs without interference from pre-miRNA and distinguished between breast cancer patients and healthy donors with an overall accuracy of 90% (Figure 3c) [58]. 

Moreover, cell culture supernatants, or EVs, in body fluids lose interaction information with other symbiotic cells in tissues, making them unable to accurately represent the role of EVs in intercellular communication [70]. Ji et al. employed spatially patterned antibody barcodes and achieved multiplexed profiling of single-cell EV secretion from over 1000 individual cells concurrently. This innovative approach enabled the comprehensive characterization of human oral squamous cell carcinoma, unraveling previously obscured single-cell heterogeneity in EV secretion dynamics (Figure 3d) [59]. This technology facilitates a thorough assessment of EV secretion diversity at the single-cell level, offering an invaluable tool to supplement existing single-cell analysis and EV research. Afterwards, they applied this platform to analyze the characteristic spectra of paired neuronal–microglial and neuronal–astrocyte single-cells in the human cell lineage. These results provide a basis for exploring how neurons and immune cells interact through complex secretion networks [71]. 

While fluorescence technology utilizing microfluidic techniques has been successfully employed in EV detection, offering commendable accuracy and high sensitivity, it is not without its drawbacks. Some issues still remain, including the intricate preparation of fluorescence labels and the occurrence of spectral interference. This interference encompasses background interference and spectral overlap issues, along with the phenomenon of photobleaching in fluorescent molecules. The resolution of these issues will contribute to enhancing the accuracy of fluorescence detection of EVs.

### 3.2. Visualization Detection

In recent years, the method of detecting EVs through microfluidic technology using the colorimetric method has undergone significant development and has simplified the equipment [34]. Visualization detection relies on the changes that can be observed by naked eyes, such as color changes within the detection system, which occur as a result of chemical or biochemical interactions between specific target analytes and colorimetric probes. One significant advantage of visualization assays is their independence from bulky off-chip detection systems. Consequently, visualization detection has garnered growing interest in biomedical research, particularly for disease diagnosis, owing to its distinct advantages in EV detection [70].

For instance, Chen et al. introduced a traditional colorimetric technique to detect EVs using a 3-D scaffold chip [72]. They proposed a ZnO nanowire-coated 3D scaffold chip device for effective immune capture and classical visual and colorimetric detection of EVs. In the work by Di et al., a rapid analysis method was introduced, utilizing nano-enzyme-assisted immunosorbent assays, eliminating the need for antibody detection [73]. The approach involved the immobilization of nanoparticles on the phospholipid membrane of exosomes, followed by the addition of chromogenic agents. For another, Jiang’s group developed a sensor platform that can visually analyze EV surface proteins in minutes. The sensor consists of a gold nanoparticle (AuNP) and a set of aptamers [74]. In addition, Ko et al. engineered a photofluidic platform powered by smartphones to quantify brain-derived exosomes. This innovative chip enables rapid processing, delivering results within one hour, which is ten-fold quicker than conventional methods. The device boasts a detection limit of approximately 10^7^ exosomes/mL [75]. Utilizing enzyme amplification, it can detect exosome biomarkers, with results easily read through a smartphone camera. 

Although many researchers have made incredible progress in the field of visualization detection methods, which can rapidly detect a wide range of biomolecules, from infectious disease-related protein biomarkers to glucose and nucleic acids, the extensive application of visualization detection is limited. It is mainly used in underdeveloped regions, and low sensitivity is its major drawback.

### 3.3. Electrochemical Detection

Electrochemical methods transfer the signals of EV recognition to electrochemical signals, such as voltage, current, and resistance [76]. Microfluidic-based electrochemical techniques have attracted great attention in EV detection due to their broad detection range, high sensitivity, and specificity.

Electrochemical methods are usually used to profile EV surface proteins. For example, Akagi et al. presented an on-chip immunoelectrophoresis method for EV protein expression analysis based on the different positive charges on the EV surface caused by antibody binding [77]. Moreover, Akagi et al. found that exosomes from different cells had differential zeta potential. Thus, they developed an electrophoresis apparatus for tracking individual exosomes [78]. Moreover, Akagi et al. developed an on-chip microcapillary electrophoresis (µCE) system to detect the zeta potential distribution of exosomes from normal cells and prostate cancer cells [79]. They found that the huge negative charge of cancer exosomes was due to abundant sialic acids. Currently, affinity ligands are usually modified on microfluidic chips for EV capture. Then, the electrochemically responsive molecules were triggered to cause a change in electrochemical signals. To date, great efforts have been devoted to introducing various signal production and amplification strategies, such as metal nanoparticles, tetrahedral DNA nanostructures (TDNs), and nucleic acid-based amplification analysis, to electrochemical biosensors for EV detection. For example, Wang et al. have developed a new filter electrochemical microfluidic chip (FEMC) that integrates on-chip separation and in situ surface protein electrochemical analysis of exosomes in the whole blood of breast cancer patients [80]. In this system, zirconium-based metal–organic frameworks (Zr-MOFs) loaded with numerous electroactive methylene blue molecules (Zr-MOFMB@UiO-66) were attached to exosomes collected on electrode surfaces, leading to the amplification of electrical signals. The entire FEMC assay took 1 h to complete, enabling timely and more informed opportunities for the diagnosis of breast cancer. To highly sensitively detect colorectal cancer exosomes, a microfluidic electrochemical biosensing platform based on TDN-based signal amplification was constructed (Figure 4a) [81]. TDNs, including the EpCAM aptamer, were immobilized on Au nanoparticles (AuNPs) as a recognition element to harvest the exosomes. Then, the AuNPs had an obvious catalytic effect on the redox reaction of ferricyanide, enabling electrochemical detection. The platform had a broad measurement range (50–10^5^ particles/µL) and a low limit of detection (42 particles/µL). 

Moreover, an increasing number of nucleic acid amplification methods have been employed in electrochemical biosensors. Xu et al. proposed a two-stage microfluidic platform (ExoPCD-chip) for the electrochemical analysis of hepatocellular exosomes in serum (Figure 4b) [82]. Particularly, exosomes captured by electrochemical aptasensors with a CD63 aptamer led to the accumulation of the hemin/G-quadruplex. This complex could function as a NADH oxidase and horseradish peroxidase (HRP)-mimicking DNAzyme simultaneously. Thus, the freshly formed H_2_O_2_ by NADH oxidation could be continuously catalyzed, accompanied by significant signal enhancement. Moreover, a staggered Y-shaped micropillar mixing pattern was introduced to create an anisotropic flow without any surface modification to improve exosome enrichment efficiency. Due to their flexible programmability, aptamers are easily engineered for signal amplification to improve EV detection sensitivity. For example, a hemin/G-quadruplex system and rolling circle amplification (RCA) were combined in an aptasensor for the selective and sensitive detection of gastric cancer exosomes (Figure 4c) [83]. RCA is recognized as a nucleic acid amplification analysis that can be performed at room temperature to preserve the integrity of exosomes. In addition, Zhang et al. designed a remarkably selective electrochemical micro-aptasensor with a detection limit of 5 × 10^2^ exosomes/mL by integrating a micropatterned electrochemical aptasensor and a signal amplification strategy of hybridization chain reaction (HCR) (Figure 4d) [84]. Biotin-labeled HCR products were used to bind specifically to enriched exosomes, utilizing EpCAM aptamers as a bridge. This was followed by the attachment of multiple avidin-HRPs, producing a current signal through the enzyme reaction. Moreover, the proposed aptasensor was effective in discriminating serum samples from early-stage lung cancer patients and late-stage patients, indicating significant promise for early cancer diagnosis.

To sum up, electrochemistry proves highly suitable for EV analysis within an integrated microfluidic chip, offering a multitude of advantages. Furthermore, no requirements for optical transparency expand the choice of materials in electrochemical response. However, contamination and changes in pH, temperature, and ionic concentration often influence the lifetimes of electrodes, which needs to be addressed [85]. So far, only a limited number of microfluidic devices incorporated with electrochemical techniques have been developed for EV detection. We believe that there will be a growing number of microfluidic devices combined with electrochemical detection as a promising means for point-of-care diagnostics.

### 3.4. Surface-Enhanced Raman Spectroscopy (SERS)

Surface-enhanced Raman spectroscopy (SERS) effectively generates spectra on certain metal surfaces, providing vibrational and rotational energy information about molecules, which is reflected in spectral peaks used to specifically identify molecules [85]. However, SERS measurements present significant challenges to reproducibility and sensitivity [86]. In this aspect, microfluidic surface-enhanced Raman spectroscopy (MF-SERS) is making progress in resolving some significant and previously insurmountable issues and limitations of SERS detection to some extent, thus improving detection capability and extending its application [87]. Through the utilization of high-throughput nanosurface microfluidics control technology and unique fingerprint identification, precise testing of ultra-small populations of biochemical particles such as cancer EVs is made possible.

In efforts to amplify Raman signals from cancer-derived EVs, Mahsa et al. developed a nanosurface fluidic device for label-free, non-immunological SERS detection of EVs. This device effectively distinguished the SERS fingerprint of EVs from noncancerous glial cells (NHA) and two subpopulations of the GBM EVs (i.e., U87 and U373). The sample solution flowed from the input ports to the serpentine analysis channels (50 × 250 μm^2^) to achieve a single-layer distribution of EVs on the nanosurface. At the same time, metal nanomaterials with SERS activity formed a hexagonal nanoscale triangular array, with each of the two triangles forming a bowtie structure with a suspended gap region area to amplify the EM field enhancement, with an electromagnetic field enhancement factor of 9 × 10^5^ (Figure 5a) [88]. In addition to the above-mentioned label-free microfluidic Raman chip, Wang et al. developed a new one with immunoassays for quickly and sensitively detecting the exosomes (Figure 3b) [89]. Hybrid channels of triangular column arrays were used to enrich CD63-positive exosomes and were fixed in the Raman detection region. EpCAM-labeled Raman beads with high densities of nitrile were used as probes for detection, and the detection limit was 1.6 × 10^2^ particles per mL with 20 μL samples. 

Various biomarkers can indicate diverse biological functions, making it significant to detect multiple exogenous biomarkers. Han et al. proposed a microfluidic-based SERS detection technique for profiling numerous exosomal biomarkers to diagnose osteosarcoma. Gold nanoparticles labeled with SERS tags can selectively bind to exosomes using specific antibodies in samples, forming exosome immunocomplexes. A microfluidic chip, comprising two symmetrical polydimethylsiloxane (PDMS) layers and a nanoporous polycarbonate track-etched (PCTE) membrane, was employed for exosome purification [91]. Microfluidic tangential flow filtration effectively eliminated plasma biomolecules and free SERS tags while enriching exosome immunocomplexes on the membrane for in situ SERS analysis [91,92]. Herein, Wang et al. also showcased a multiplex EV phenotype analyzer chip (EPAC). EPAC integrates a nanomixing-enhanced microchip and a multiplex surface-enhanced Raman scattering (SERS) nanotag system for direct EV phenotyping (Figure 5c) [90]. They observed the EV phenotypic heterogeneity and longitudinally monitored the EV phenotypic evolution, finding specific EV profiles involved in the development of drug resistance and the potential of EV phenotyping for monitoring treatment responses. Thus, the microfluidic-based SERS detection method offers great potential for the detection of external vesicles and cancer diagnosis. 

## 4. Conclusions and Perspectives

EVs are intricately linked to numerous physiological processes as well as the onset and progression of diseases. Efficient EV isolation methods and sensitive EV detection methods will help to improve the understanding of the physiological and pathological effects of EVs and provide important support for the precision medicine of related diseases. At present, conventional EV isolation and detection technologies still have limitations. Microfluidic-based methods for isolation and detection of EVs have the obvious advantages of high integration, low consumption, fast speed, high separation efficiency, and high detection sensitivity, which opens up new ideas and directions for the research of EVs. With the help of microfluidic chips, the efficient isolation, enrichment, and multi-marker detection of EVs with different sizes can be integrated into a single chip, and a more diversified clinical detection instrument can be built.

The field of microfluidic-based EV isolation and detection is still in its infancy, and there are still a lot of theoretical and technical problems to be solved. Therefore, the means to achieve highly selective and accurate isolation of EVs in actual biological samples, sensitive and selective detection of these EVs, and even the biological information carried in them will be important topics in the study of EVs based on microfluidic chips. With the development of technology and in-depth research, the isolation and analysis of individual EVs can be realized by the microfluidic method, and commercial EV chips are also expected to be applied in clinical practice.

In this article, the existing methods of microfluidic-based EV isolation and analysis are reviewed. Compared with traditional ultracentrifugation, ultrafiltration, immunocapture, and co-precipitation, microfluidic chips are smaller and more flexible, and microfluidic immunoaffinity methods can isolate high-purity EVs with strong specificity. The isolation methods based on the physical characteristics of EVs do not need to add expensive reagents such as antigens and antibodies. Therefore, the cost is low, and the isolation process will not cause contamination, which is conducive to downstream analysis. Microfluidic-based EV analysis methods have the advantages of fast analysis speed, high throughput, and low reagent consumption, which can meet the needs of rapid detection of EVs in a large number of clinical samples. Therefore, microfluidic technology has significant advantages in the isolation of EVs in a small number of clinical samples and the rapid estimation of diseases.

Despite the significant advancements in microfluidics-based capturing and detection technologies for EV analysis, several challenges remain to be addressed. One of the primary challenges is the standardization of isolation and detection protocols to ensure consistency and reliability across different platforms and laboratories. Variability in sample preparation, device design, and operating conditions can lead to inconsistent results and hinder the reproducibility of findings. Therefore, efforts are needed to establish standardized protocols and quality control measures to facilitate the comparison and validation of results obtained from different microfluidic-based platforms.

Another challenge is the optimization of microfluidic devices for the analysis of specific EV subpopulations or cargo molecules. EVs exhibit heterogeneity in size, surface markers, and cargo content, which necessitates the development of tailored microfluidic devices capable of selectively capturing and analyzing desired EV subtypes. Furthermore, the integration of multiplexed detection modalities into microfluidic platforms would enable comprehensive profiling of EVs, facilitating the discovery of novel biomarkers and the elucidation of disease mechanisms.

In addition to technical challenges, the translation of microfluidic-based EV analysis from research laboratories to clinical settings requires overcoming regulatory and commercialization hurdles. Regulatory agencies, such as the Food and Drug Administration (FDA), require rigorous validation of diagnostic assays and devices to ensure their safety and efficacy for clinical use. Moreover, the scalability and cost-effectiveness of microfluidic-based platforms need to be optimized to enable widespread adoption in clinical diagnostics and personalized medicine.

In recent years, with the rapid development of micro/nano manufacturing, new materials, and information technology, the design of microfluidic chips and the performance of supporting devices have been further improved. It is mainly reflected in (1) the development of precision manufacturing technology, making it possible to integrate multiple EV isolation methods and realize the integration of EV isolation and detection on one chip; (2) by combining the chips and portable detection equipment, a miniaturized EV microfluidic isolation and analysis platform is constructed to realize the rapid detection of EVs and greatly expand its application space. With the miniaturization, integration, and automation of microfluidic EV isolation and analysis devices, microfluidic technology will play an increasingly important role in EV isolation, biochemical detection, and mechanism research.

In conclusion, microfluidics-based capturing and detection technologies offer powerful tools for the isolation, characterization, and analysis of extracellular vesicles. These technologies leverage precise fluid manipulation at the microscale level to enable rapid, efficient, and high-throughput analysis of EVs from complex biological samples. Despite remaining challenges in standardization, optimization, and translation to clinical applications, ongoing research efforts are poised to overcome these hurdles and unlock the full potential of microfluidic-based EV analysis in disease diagnosis, prognosis, and therapeutics. With continued innovation and collaboration across interdisciplinary fields, microfluidics-based EV analysis holds promise for revolutionizing personalized medicine and improving patient outcomes.

## Figures and Tables

**Figure 1 micromachines-15-00630-f001:**
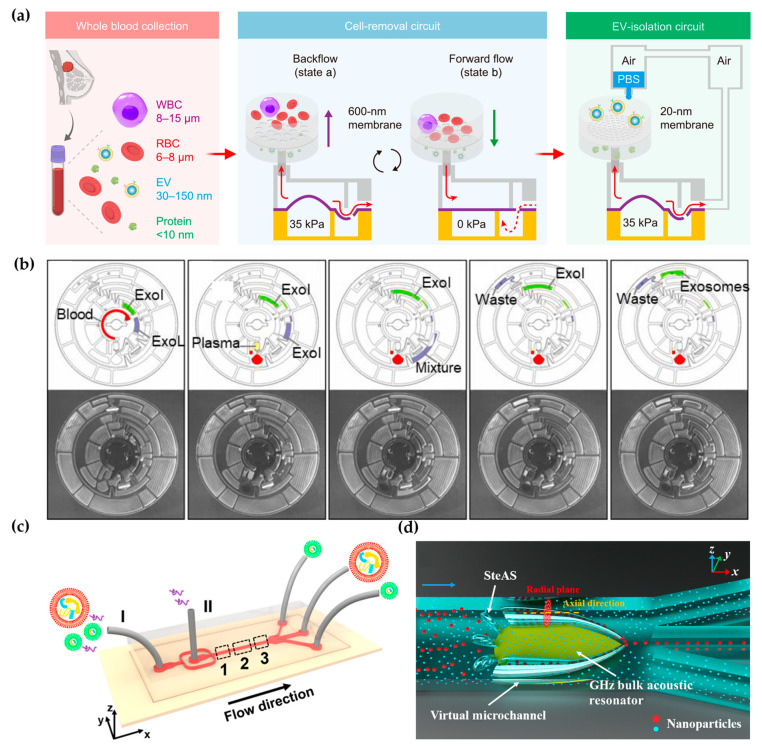
Label-free microfluidic-based EV isolation methods: (**a**) Cascaded microfluidic circuits for EV isolation. Reproduced from Ref. [23]. (**b**) Centrifugal microfluidic disc for EV enrichment. Reproduced from Ref. [24]. (**c**) Visual representation of the separation mechanism of viscoelastic microfluidics. I: inlet for sample fluids, II: inlet for sheath fluids. Reproduced from Ref. [26]. (**d**) Mechanism of the stereo acoustic stream (SteAS) platform. Reproduced from Ref. [27].

**Figure 2 micromachines-15-00630-f002:**
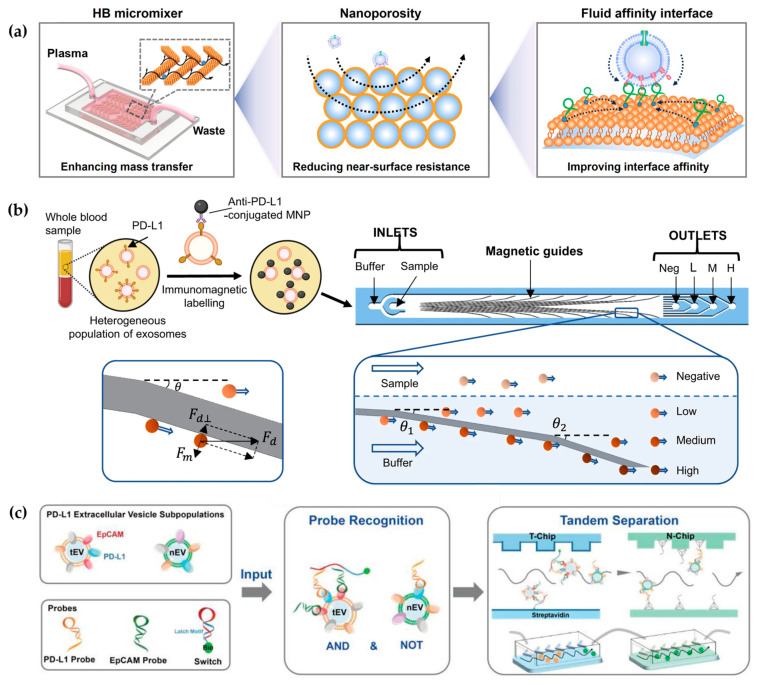
Affinity-based microfluidic EV isolation methods: (**a**) The schematic diagram of the FluidporeFace chip. Reproduced from Ref. [41]. (**b**) A magnetic deflection-based NanoEPIC system to achieve phenotypic profiling and nanoscale sorting of sEVs. Reproduced from Ref. [45]. (**c**) DNA computation-mediated microfluidic tandem separation for PD-L1^+^ EV subpopulations. Reproduced from Ref. [46].

**Figure 3 micromachines-15-00630-f003:**
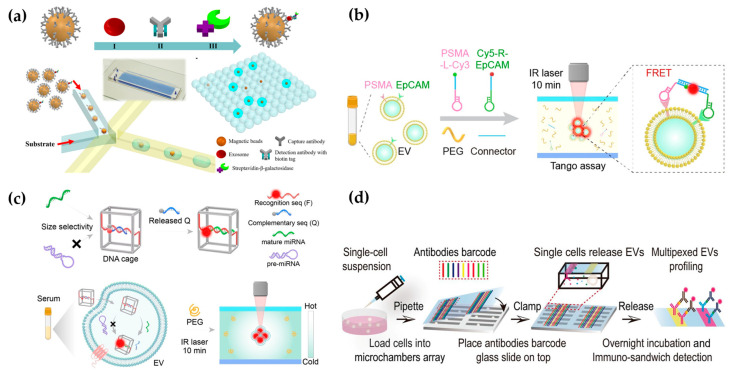
Microfluidic-based fluorescent detection: (**a**) The droplet digital ExoELISA method for quantifying EVs. Reproduced from Ref. [56]. (**b**) Schematic of the one-step thermophoretic AND gate operation (Tango) assay. Reproduced from Ref. [57]. (**c**) DTTA for detecting mRNA within EVs. Reproduced from Ref. [58]. (**d**) Schematic illustration depicting the workflow for the multiplexed profiling of single-cell EV secretion. Reproduced from Ref. [59].

**Figure 4 micromachines-15-00630-f004:**
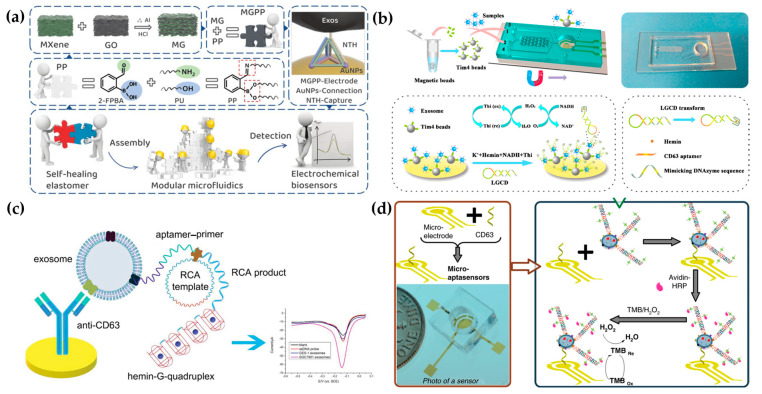
Microfluidic-based electrochemical detection: (**a**) TDN-based microfluidic electrochemical biosensing platform for exosome analysis. Reproduced from Ref. [81]. (**b**) Schematic diagram of the ExoPCD chip. Reproduced from Ref. [82]. (**c**) Hemin/G-quadruplex-assisted signal amplification. Reproduced from Ref. [83]. (**d**) Electrochemical micro-aptasensors based on HCR for exosome detection. Reproduced from Ref. [84].

**Figure 5 micromachines-15-00630-f005:**
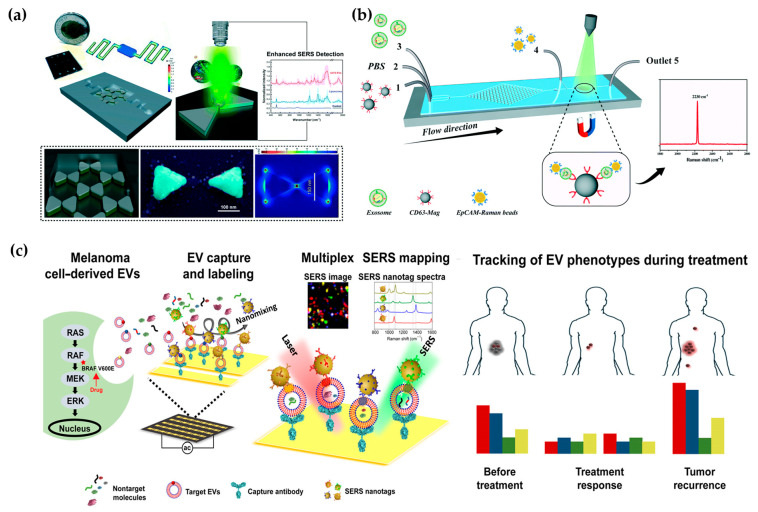
Microfluidic-based SERS detection. (**a**) Spatially suspended nanobowtie surface microfluidic device for SERS detection of EVs. Reproduced from Ref. [88]. (**b**) Microfluidic Raman chip for exosome capture and detection. Reproduced from Ref. [89]. (**c**) EV phenotyping by EPAC. Four colors represent four biomarkers: MCSP-MBA, red; MCAM-TFMBA, blue; ErbB3-DTNB, green; LNGFR-MPY, yellow. Reproduced from Ref. [90].

**Table 1 micromachines-15-00630-t001:** Traditional techniques versus microfluidic-based methodologies for EV analysis [49,50,51].

Aspect	Traditional Techniques	Microfluidic-Based Methodologies
Effectiveness	Varied, depending on method (e.g., ultracentrifugation, precipitation)	High, with precise control over fluid manipulation and surface interactions
Efficiency	Time-consuming, labor-intensive	Rapid, automated processes with minimal sample and reagent consumption
Practicality	Limited scalability, manual operation	Scalable, integrated systems suitable for high-throughput analysis
Specificity	May lack specificity, leading to contamination and low yield	Enhanced specificity, with tailored devices for selective EV capture
Reproducibility	Variable due to manual handling and batch-to-batch variability	Improved reproducibility with standardized protocols and automated workflows

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
