# Peer review of "Recent Advances in Microfluidic-Based Extracellular Vesicle Analysis"

_micromachines, 2024, doi:10.3390/mi15050630_

Round 1

Reviewer 1 Report

Comments and Suggestions for Authors

I am pleased to review the manuscript tilted as “Recent advances in microfluidic-based extracellular vesicle Analysis”. Review article summarizes the traditional techniques alongside microfluidic-based methodologies for the isolation and detection of EVs. Highlight advantages of microfluidic technology in enhancing EV capture efficiency and targeted detection with prospects for achieving automated and high-throughput EV detection in clinical samples.

I do believe it’s a well written and well-constructed manuscript, showing the authors hard work and dedication. However, based on my review of the manuscript I would like to suggest some of the points for the overall improvement of the manuscript.

Following are my comments that I would like to be taken into consideration by the authors prior to getting my green signal to endorse the publication of this manuscript.

1.     In abstract line 11-13. Please rewrite this as it’s hard to understand and very important lines to open up the topic of the paper. Write it simple and clear.

2.     In abstract line 21-23. Very important information but hard to read please rewrite making shorter sentences instead of one long sentence for better reading and understanding.

3.     Introduction section line 41-42. Please write that in terms of microliters instead of liters.  

4.     In your review of traditional techniques and microfluidic-based methodologies, do you compare their effectiveness, efficiency, and practicality? If so, could this be presented in the form of a table?

5.     I would like you to add one more paragraph in introduction section that specifically highlights the high throughput capability and standardization of microfluidics-based capturing and detection technologies for EVs.

6.     Heading 3 line 191 Micofuldic-based fluorescent detection, should be rewritten as Micofuldic-based EVs detection.

7.     Heading 3.2-line 285 Visualization detection. Can not this be replaced with colorimetric detection for the purpose of simplicity?

8.     Please add expanded section highlighting the challenges and future perspectives followed by conclusion section.

Best wishes….

Comments on the Quality of English Language

Good

Author Response

Reply to Reviewer 1:

Point 1. In abstract line 11-13. Please rewrite this as it’s hard to understand and very important lines to open up the topic of the paper. Write it simple and clear.

Reply 1: Thanks for your kind suggestion. In the revised article, we have revised the sentence to make it easier to understand for readers.

Point 2. In abstract line 21-23. Very important information but hard to read please rewrite making shorter sentences instead of one long sentence for better reading and understanding.

Reply 2: Thanks for your kind suggestion. In the revised article, we have revised the sentence to make it more logical and easier to understand for readers.

Point 3. Introduction section line 41-42. Please write that in terms of microliters instead of liters.

Reply 3: Thanks for your kind suggestion. In the revised article, we have corrected it.

Point 4. In your review of traditional techniques and microfluidic-based methodologies, do you compare their effectiveness, efficiency, and practicality? If so, could this be presented in the form of a table?

Reply 4: Thanks for your kind advice. We have summarized the table on traditional techniques versus microfluidic-based methodologies for EV analysis. (Line 242-243)

Point 5. I would like you to add one more paragraph in introduction section that specifically highlights the high throughput capability and standardization of microfluidics-based capturing and detection technologies for EVs.

Reply 5: Thanks for your kind suggestion. According to your advice, we give more introduction in the high throughput capability and standardization of microfluidics-based capturing and detection technologies for EVs. (Line 48-63)

Point 6. Heading 3 line 191 Micofuldic-based fluorescent detection, should be rewritten as Micofuldic-based EVs detection.

Reply 6: Thanks for your kind suggestion. In the revised article, we have corrected it.

Point 7. Heading 3.2-line 285 Visualization detection. Can not this be replaced with colorimetric detection for the purpose of simplicity?

Reply 7: Thanks for your kind suggestion. Here, we make an explain about this issue. The method described in our paper differs from traditional colorimetric techniques, we think ‘visualization detection’ will be more accurate.

Point 8. Please add expanded section highlighting the challenges and future perspectives followed by conclusion section.

Reply 8: Thanks for your kind advice. In the revised article, the results section has been enhanced to focus on expanding the challenges and prospects for the future of EV separation and detection based on microfluidic technology. (Line 541-563 and 575-584)

Reviewer 2 Report

Comments and Suggestions for Authors

Authors have conducted a review of recent advances in microfluidic-based extracellular vesicle analysis but have omitted many microfluidic platforms.

Here are some suggestions to improve the manuscript further:

As a one-step method, Exo-CMDS enriched exosomes with optimal exosomal concentration of 5.1×109 particles/mL from small amount of blood samples (<300 μL) in 8 min.

The authors have not touched immunoaffinity chromatographic method with the monoliths with pore sizes of 1 µm that provide convective transport through the channels. This method can process 250 mL of plasma in under 10 min and produces exosomal concentration of 1.17 ± 0.41 * 10^12, which is 1000 more concentrated than on step method suggested above. This is also one step method, with no need for tedious microfluidic chip preparation. The monolithic column is commercially available Find more in:

Multia, Evgen, et al. "Fast isolation of highly specific population of platelet-derived extracellular vesicles from blood plasma by affinity monolithic column, immobilized with anti-human CD61 antibody." Analytica chimica acta 1091 (2019): 160-168.

lines 104-110 Monolithic column chromatography should be discussed in affinity-based EV isolation. It doesn't have the same issues with the surface, since the polymer it is made of is biocompatible. It has micrometer-sized channels that are excellent for EVs and might be a better option compared to conventional microfluidics. The monolithic disk is smaller (only 0.34 ml) than many microfluidic devices.

Authors are missing microfluidic immunocapture:

Shao, Huilin, et al. "Chip-based analysis of exosomal mRNA mediating drug resistance in glioblastoma." Nature communications 6.1 (2015): 6999.

line 272. revise the sentence: Some ons include the intricate preparation of fluorescence labels and the occurrence of spectral interference.

The authors are missing many microfluidic platforms:

Liang, Li-Guo, et al. "An integrated double-filtration microfluidic device for isolation, enrichment and quantification of urinary extracellular vesicles for detection of bladder cancer." Scientific reports 7.1 (2017): 46224.

Deterministic lateral displacement (DLD) pillar arrays

Zeming, Kerwin Kwek, et al. "Fluorescent label-free quantitative detection of nano-sized bioparticles using a pillar array." Nature communications 9.1 (2018): 1254.

Wunsch, Benjamin H., et al. "Nanoscale lateral displacement arrays for the separation of exosomes and colloids down to 20 nm." Nature nanotechnology 11.11 (2016): 936-940.

Hattori, Yuya, et al. "Micro-and nanopillar chips for continuous separation of extracellular vesicles." Analytical chemistry 91.10 (2019): 6514-6521.

Smith, Joshua T., et al. "Integrated nanoscale deterministic lateral displacement arrays for separation of extracellular vesicles from clinically-relevant volumes of biological samples." Lab on a Chip 18.24 (2018): 3913-3925.

Oscillatory viscoelastic microfluidic system

Asghari, Mohammad, et al. "Oscillatory viscoelastic microfluidics for efficient focusing and separation of nanoscale species." ACS nano 14.1 (2019): 422-433.

Also, on-chip immunoelectrophoresis and on-chip microcapillary electrophoresis of extracellular vesicles are missing. 

Akagi, Takanori, et al. "On-chip immunoelectrophoresis of extracellular vesicles released from human breast cancer cells." PloS one 10.4 (2015): e0123603.

Kato, Kei, et al. "Electrokinetic evaluation of individual exosomes by on-chip microcapillary electrophoresis with laser dark-field microscopy." Japanese Journal of Applied Physics 52.6S (2013): 06GK10.

Akagi, Takanori, et al. "Evaluation of desialylation effect on zeta potential of extracellular vesicles secreted from human prostate cancer cells by on-chip microcapillary electrophoresis." Japanese Journal of Applied Physics 53.6S (2014): 06JL01.

In addition, simultaneous isolation and preconcentration of exosomes by ion concentration polarization.

Find more information on isolation platforms:

Liangsupree, Thanaporn, et al.. "Modern isolation and separation techniques for extracellular vesicles." Journal of Chromatography A 1636 (2021): 461773. 

These examples are just scratching the surface, so the paper needs to be properly expanded and also include more recent advances. 

Comments on the Quality of English Language

English language could be improved, but overall the text is well readable.

Author Response

Point 1. As a one-step method, Exo-CMDS enriched exosomes with optimal exosomal concentration of 5.1×109 particles/mL from small amount of blood samples (<300 μL) in 8 min.

The authors have not touched immunoaffinity chromatographic method with the monoliths with pore sizes of 1 µm that provide convective transport through the channels. This method can process 250 mL of plasma in under 10 min and produces exosomal concentration of 1.17 ± 0.41 * 10^12, which is 1000 more concentrated than on step method suggested above. This is also one step method, with no need for tedious microfluidic chip preparation. The monolithic column is commercially available Find more in:

Multia, Evgen, et al. "Fast isolation of highly specific population of platelet-derived extracellular vesicles from blood plasma by affinity monolithic column, immobilized with anti-human CD61 antibody." Analytica chimica acta 1091 (2019): 160-168.

Reply 1: Thanks for your kind advice. In the revised article, We have included this method in section “2.2 Affinity-based EV isolation” (Line 148) , ref [35].

Point 2. lines 104-110 Monolithic column chromatography should be discussed in affinity-based EV isolation. It doesn't have the same issues with the surface, since the polymer it is made of is biocompatible. It has micrometer-sized channels that are excellent for EVs and might be a better option compared to conventional microfluidics. The monolithic disk is smaller (only 0.34 ml) than many microfluidic devices. 

Reply 2: Thanks for your kind advice. After carefully scrutinizing the pertinent literature as per your guidance, we have found merit in the method you proffered. Nevertheless, given the predominant focus of our article on microfluidic technology, we opine that an exhaustive exposition may be superfluous. We extend our gratitude for your invaluable counsel.

Point 3. Authors are missing microfluidic immunocapture:

Shao, Huilin, et al. "Chip-based analysis of exosomal mRNA mediating drug resistance in glioblastoma." Nature communications 6.1 (2015): 6999.

Reply 3: Thanks for your kind advice. In the revised article, we have attached this work in section “3.1 Fluorescent detection” (Line 309-311), ref [65] .

Point 4. line 272. revise the sentence: Some ons include the intricate preparation of fluorescence labels and the occurrence of spectral interference.

Reply 4: Thanks for your kind advice. We have revised “Some ons include the intricate preparation of fluorescence labels and the occurrence of spectral interference” to be “Some issues still remain, including the intricate preparation of fluorescence labels and the occurrence of spectral interference”, which can be found in the revised manuscript (line 338-339).

Point 5. The authors are missing many microfluidic platforms:

Liang, Li-Guo, et al. "An integrated double-filtration microfluidic device for isolation, enrichment and quantification of urinary extracellular vesicles for detection of bladder cancer." Scientific reports 7.1 (2017): 46224.

Deterministic lateral displacement (DLD) pillar arrays

Zeming, Kerwin Kwek, et al. "Fluorescent label-free quantitative detection of nano-sized bioparticles using a pillar array." Nature communications 9.1 (2018): 1254.

Wunsch, Benjamin H., et al. "Nanoscale lateral displacement arrays for the separation of exosomes and colloids down to 20 nm." Nature nanotechnology 11.11 (2016): 936-940.

Hattori, Yuya, et al. "Micro-and nanopillar chips for continuous separation of extracellular vesicles." Analytical chemistry 91.10 (2019): 6514-6521.

Smith, Joshua T., et al. "Integrated nanoscale deterministic lateral displacement arrays for separation of extracellular vesicles from clinically-relevant volumes of biological samples." Lab on a Chip 18.24 (2018): 3913-3925.

Oscillatory viscoelastic microfluidic system

Asghari, Mohammad, et al. "Oscillatory viscoelastic microfluidics for efficient focusing and separation of nanoscale species." ACS nano 14.1 (2019): 422-433.

Also, on-chip immunoelectrophoresis and on-chip microcapillary electrophoresis of extracellular vesicles are missing. 

Akagi, Takanori, et al. "On-chip immunoelectrophoresis of extracellular vesicles released from human breast cancer cells." PloS one 10.4 (2015): e0123603.

Kato, Kei, et al. "Electrokinetic evaluation of individual exosomes by on-chip microcapillary electrophoresis with laser dark-field microscopy." Japanese Journal of Applied Physics 52.6S (2013): 06GK10.

Akagi, Takanori, et al. "Evaluation of desialylation effect on zeta potential of extracellular vesicles secreted from human prostate cancer cells by on-chip microcapillary electrophoresis." Japanese Journal of Applied Physics 53.6S (2014): 06JL01.

In addition, simultaneous isolation and preconcentration of exosomes by ion concentration polarization.

Find more information on isolation platforms:

Liangsupree, Thanaporn, et al.. "Modern isolation and separation techniques for extracellular vesicles." Journal of Chromatography A 1636 (2021): 461773. 

These examples are just scratching the surface, so the paper needs to be properly expanded and also include more recent advances. 

Reply 5: Thanks for your kind advice. In the revised article, we have attached the examples in ref 22, 29, 30, 32, 31, 25, 79, 80, 81, included more recent advances and marked the revisions in red.

Reviewer 3 Report

Comments and Suggestions for Authors

The authors comprehensively reviewed the recent advancements in microfluidic-based isolation and detection of extracellular vesicles (EVs). EVs have been demonstrated as essential messengers that mediate intercellular communication. This review underscored its transformative impact on diagnosing and treating diseases. The authors highlighted the limitations of traditional EV isolation and detection and detailed various advanced microfluidic-based isolation techniques, ranging from label-free to affinity-based separation methods. In addition, the review delves into microfluidic-based detection techniques like fluorescence, electrochemical detection, and surface-enhanced Raman scattering, which underscores the technology's potential for precise and efficient EV analysis. The review also addresses the current challenges and future prospects in this evolving field. I found this article very interesting and informative, and I recommend publishing it after minor revisions:

1.     In the concluding section of “2.2 Affinity-Based Isolations” (Lines 175-182), the authors point out that certain limitations require the innovation and refinement of these techniques. However, they do not specify the drawbacks of affinity-based EV isolation methods. It would be beneficial for the authors to elaborate on these limitations in greater detail to provide clarity on this aspect of this methodology.

2.    I recommend removing the word “fluorescent” from the title of Section 3 “microfluidic-based fluorescent detection” to prevent confusion with the subtitle of Section 3.1, “Fluorescent Detection.”

3.     There are a few citation/reference errors in the last part of the manuscript (sections 3.3 and 3.4) that should be corrected:

a.     Line 338, please delete the inserted citation “[Microfluidic Technology for the Isolation and Analysis of Exosomes]”

b.     Reference 72, mentioned on Line 348, does not correspond to any citation within the text. Please consider deleting it.

c.      Line 414, the correct reference number is 82, not 81

d.     Line 419, the citation note “[81][62]” should be corrected to reference number 83

e.     The reference numbers in Figure 5 are all incorrect. The [78] should be changed to [79], [79] to [80], and [82] to [83].

4.     Some of the reference formats are incorrect (missing volume and page numbers), making it difficult for readers to locate the original articles. For example, 52, 53, 54, 56, 57, 59, etc.

Comments on the Quality of English Language

Generally, the English writing in this article is fluent and readable. I recommend that the author proofread the manuscript and correct a few issues before publishing:

1.     Spell issue in line 91 “applicaiontion” should be “application.”

2.     In lines 83-86, the sentence: “a gigahertz bulk acoustic resonator and microfluidics triggered and stabilized acoustic waves and streams to form a virtual channel with a self-adjusted diameter from dozens to a few micrometers” is difficult to understand clearly. I suggest revising it to “A gigahertz bulk acoustic resonator, in combination with microfluidics, triggers and stabilizes acoustic waves and streams. This process forms a virtual channel whose diameter can self-adjust, ranging from dozens to a few micrometers.”

3.     Line 272, “some ons” should be revised to “some issues.”

Author Response

Point 1. In the concluding section of “2.2 Affinity-Based Isolations” (Lines 175-182), the authors point out that certain limitations require the innovation and refinement of these techniques. However, they do not specify the drawbacks of affinity-based EV isolation methods. It would be beneficial for the authors to elaborate on these limitations in greater detail to provide clarity on this aspect of this methodology.

Reply 1: Thanks for your kind advice. In the revised article, we have expanded the drawbacks of affinity-based EV isolation methods. (Line 225-233)

Point 2. I recommend removing the word “fluorescent” from the title of Section 3 “microfluidic-based fluorescent detection” to prevent confusion with the subtitle of Section 3.1, “Fluorescent Detection.”

Reply 2: Thanks for your kind advice. In the revised article, we have corrected it.

Point 3. There are a few citation/reference errors in the last part of the manuscript (sections 3.3 and 3.4) that should be corrected:

  1. Line 338, please delete the inserted citation “[Microfluidic Technology for the Isolation and Analysis of Exosomes]”
  2. Reference 72, mentioned on Line 348, does not correspond to any citation within the text. Please consider deleting it.
  3. Line 414, the correct reference number is 82, not 81
  4. Line 419, the citation note “[81][62]” should be corrected to reference number 83
  5. The reference numbers in Figure 5 are all incorrect. The [78] should be changed to [79], [79] to [80], and [82] to [83].

Reply 3: Thanks for your kind advice. In the revised article, we have checked the reference and corrected it in the full text carefully.

Point 4. Some of the reference formats are incorrect (missing volume and page numbers), making it difficult for readers to locate the original articles. For example, 52, 53, 54, 56, 57, 59, etc.

  1. Spell issue in line 91 “applicaiontion” should be “application.”
  2. In lines 83-86, the sentence: “a gigahertz bulk acoustic resonator and microfluidics triggered and stabilized acoustic waves and streams to form a virtual channel with a self-adjusted diameter from dozens to a few micrometers” is difficult to understand clearly. I suggest revising it to “A gigahertz bulk acoustic resonator, in combination with microfluidics, triggers and stabilizes acoustic waves and streams. This process forms a virtual channel whose diameter can self-adjust, ranging from dozens to a few micrometers.”
  3. Line 272, “some ons” should be revised to “some issues.”

Reply 4: Thanks for your kind advice. In the revised article, we have checked the grammar and spelling in the full text carefully to make it more logical and easier to understand for reader.

Round 2

Reviewer 1 Report

Comments and Suggestions for Authors

I am satisfied with the kind response of the authors to my suggestions. I agree to the publication of this article in its present form. Thank you. 

Reviewer 2 Report

Comments and Suggestions for Authors

The authors have adequately replied to the comments.

Reviewer 3 Report

Comments and Suggestions for Authors

There is no issue after revision. I recommend accepting the present form.